**Data Availability Statement:** The data studied and R code are available on the Pediatric Sepsis CoLab Dataverse (https://doi.org/10.5683/SP3/MSTH98), subject to an application meeting the ethical and

# A proposed de-identification framework for a cohort of children presenting at a health facility in Uganda

Alishah Mawji[1,2]*, Holly Longstaff[3], Jessica Trawin[2], Dustin Dunsmuir[1,2], Clare Komugisha[4], Stefanie K. Novakowski[1], Matthew O. Wiens[2,4,5], Samuel Akech[6], Abner Tagoola[7], Niranjan Kissoon[2,8], J. Mark Ansermino[1,2]

1 Department of Anesthesiology, Pharmacology & Therapeutics, University of British Columbia, Vancouver, British Columbia, Canada, 2 Centre for International Child Health, BC Children's Hospital Research Institute, Vancouver, British Columbia, Canada, 3 Privacy and Access, PHSA Research and New Initiatives, Provincial Health Services Authority, Vancouver, British Columbia Canada, 4 WALIMU, Kololo, Kampala, Uganda, 5 Mbarara University of Science and Technology, Mbarara, Uganda, 6 Kenya Medical Research Institute/ Wellcome Trust Research Programme, Nairobi, Kenya, 7 Department of Pediatrics, Jinja Regional Referral Hospital, Rotary Rd, Jinja, Uganda, 8 Department of Pediatrics, University of British Columbia, Vancouver, British Columbia, Canada

* alishah.mawji@bcchr.ca

## Abstract

Data sharing has enormous potential to accelerate and improve the accuracy of research, strengthen collaborations, and restore trust in the clinical research enterprise. Nevertheless, there remains reluctancy to openly share raw data sets, in part due to concerns regarding research participant confidentiality and privacy. Statistical data de-identification is an approach that can be used to preserve privacy and facilitate open data sharing. We have proposed a standardized framework for the de-identification of data generated from cohort studies in children in a low-and-middle income country. We applied a standardized de-identification framework to a data sets comprised of 241 health related variables collected from a cohort of 1750 children with acute infections from Jinja Regional Referral Hospital in Eastern Uganda. Variables were labeled as direct and quasi-identifiers based on conditions of replicability, distinguishability, and knowability with consensus from two independent evaluators. Direct identifiers were removed from the data sets, while a statistical risk-based de-identification approach using the k-anonymity model was applied to quasi-identifiers. Qualitative assessment of the level of privacy invasion associated with data set disclosure was used to determine an acceptable re-identification risk threshold, and corresponding k-anonymity requirement. A de-identification model using generalization, followed by suppression was applied using a logical stepwise approach to achieve k-anonymity. The utility of the de-identified data was demonstrated using a typical clinical regression example. The de-identified data sets was published on the Pediatric Sepsis Data CoLaboratory Dataverse which provides moderated data access. Researchers are faced with many challenges when providing access to clinical data. We provide a standardized de-identification framework that can be adapted and refined based on specific context and risks. This process will be combined with moderated access to foster coordination and collaboration in the clinical research community.

governance requirements of the CoLab (contact jessica.trawin@cw.bc.ca).

**Funding:** This research was supported by the Wellcome Trust Innovator Award (ID: 215695/B/19/Z), awarded to JMA and SA. The funders of the study had no role in study design, data collection, data analysis, data interpretation, or writing of the manuscript.

**Competing interests:** JMA serves as a section editor for PLOS Digital Health. The peer-review process was guided by an independent editor, and the authors have no other competing interests to declare.

## Author summary

Open Data is data that anyone can access, use, and share. Open Data has the potential to facilitate collaboration, enrich research, and advance the analytic capacity to inform decisions. Importantly, Open Data plays a role in fulfilling obligations to research participants and honoring the nature of medical research as a public good. Leaders in industry, academia, and regulatory agencies recognize the value in increased transparency and are focusing on how to openly share data while minimizing the safety risks to research participants. For example, making data open can pose a privacy risk to research participants who have shared personal health information. This risk can be mitigated using data de-identification, a process of removing personal information from a data sets so that an individual's identity is no longer apparent or cannot be reasonably ascertained from the data. We introduce a simple, statistical risk-based framework for de-identification of clinical data that can be followed by any researcher. This framework will guide open data sharing while improving the protection of research participants.

## Introduction

There are increasing requirements from governments, funders, publishers, and patients to make clinical research data more widely accessible, as Open Data [1,2,3,4]. The benefits of Open Data are manifold and include the following: ensures study results are both transparent and verifiable, enables low-cost secondary analysis of the data, encourages collaboration in the research community, and enhances public confidence in the scientific process [5,6,7]. This prompted several new initiatives directed to assisting individual researchers to make clinical data available, such as The Dataverse Project, Dryad, Vivli and The Yoda Project. [8,9,10]. Despite the myriad of advantages and emergence of open data sharing platforms, most raw data sets are not openly shared [11]. Major concerns about making data open include the risk of breaching participant privacy or producing new unanticipated harms, such as stigmatization, to individuals or entire groups of participants [12,13].

These concerns can be mitigated by implementing data de-identification, a process of removing personal information from a data sets such that disclosure does not violate the privacy of individuals [14]. The Information and Privacy Commissioner of Ontario and the Health Insurance Portability and Accountability Act (HIPAA) provide guidance to data de-identification and suggest that variables are to be handled according to classification as direct, or quasi-identifiers. Direct identifiers, those variables that can uniquely identify an individual, should be completely removed from the de-identified data sets [14,15]. Quasi-identifiers are variables that (1) an adversary is assumed to have background knowledge of, and (2) can be used either individually, or in combination to re-identify an individual in the data set [14]. For these variables, the risk of re-identification must be weighed against the perceived benefit of the use of the data. This process can be facilitated by statistical methods to estimate the risk, and de-identification techniques to reduce the risk to an acceptable level.

A prominent model to evaluate re-identification risk is k-anonymity [16]. This model suggests the probability of re-identifying a record to be the reciprocal of the number of records in the data set with identical values across quasi-identifiers. A data set is k-anonymous if the quasi-identifiers for each record are indistinguishable from k-1 other records in the data set. Two common techniques that can be applied to achieve k-anonymity are generalization and suppression [16]. Generalization occurs when attribute values for a variable are combined to

create a broader category that will contain more records. Generalization is performed with user-defined hierarchies, which are transformation rules that reduce the precision of attribute values in a stepwise manner. Suppression occurs when values that violate anonymity standards are deleted from the data set entirely. Application of generalization, followed by suppression, has been recommended for the biomedical domain [16].

While there exist general guidelines for data de-identification, many researchers lack the knowledge for effective application to a clinical data sets and there are no readily available resources to assist this process. We describe a framework for data de-identification using a clinical example and step-by-step instructions that can be followed by any researcher looking to comply with funder or publisher requests to make the data open.

## Methods

De-identification of the Smart Triage data sets followed a six-step framework (Fig 1). Functional definitions of key terms used in the framework described are outlined in the S1 Appendix.

### Data sets

The Smart Triage data sets was generated from a prospective cohort study conducted between April 2020 and March 2021 at Jinja Regional Referral Hospital in Jinja, Uganda [17,18]. The study was reviewed by the institutional review boards at the University of British Columbia in Canada (ID: H19-02398; H20-00484), the Makerere University School of Public Health in Uganda and the Uganda National Council for Science and Technology. Children and youth under 19 years of age seeking treatment for an acute illness at the pediatric outpatient department between 8:00 am and 5:00 pm were eligible. Participation was voluntary and written informed consent was provided by a parent or guardian prior to enrollment. Assent was required from children above eight years of age. Consent was obtained to make de-identified data available to other researchers. There were 241 health-related variables, including clinical signs and symptoms, and anthropometric and sociodemographic information, which were collected from 1764 participants (S2 Appendix). This data sets was generated to inform development of a rapid triage model for children presenting to the emergency department at health facilities in low-and-middle income countries.

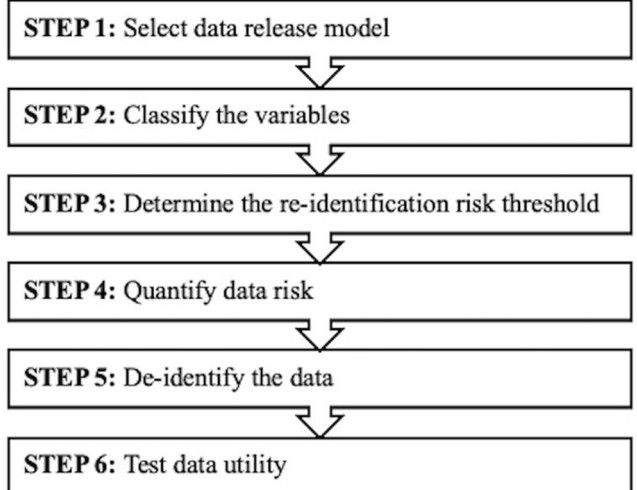

**Fig 1. Six-step de-identification framework.**

## Step 1: Select data release model

A de-identified data sets may be released publicly, semi-publicly, or non-publicly [14]. The release model plays an important role in determining the amount of de-identification required as certain models offer more privacy protection than others.

In a public data release, the data sets is available for anyone to download or use without any conditions [14]. This model provides the greatest availability and least amount of protection. On the other hand, a non-public data release limits data sets availability to a select number of recipients. As a condition of receiving the data, recipients must agree to terms and conditions regarding the privacy and security of the data. This model provides the least availability and highest amount of protection. In a semi-public data release, the data set is available to anyone for download; however, as a condition of receiving the data, the recipient must register with the organization releasing the data set and agree to the restrictions regarding the processing and sharing of data.

We have selected a semi-public release for the de-identified Smart Triage data sets to be published on the Pediatric Sepsis Data CoLaboratory (Sepsis CoLab) Dataverse, a platform that allows for international data sharing among members with built-in access control [19]. Data collaborators must register with the CoLab and sign a memorandum of understanding, submit a project proposal detailing what they plan to do with the data, and sign a terms-of-use agreement.

Selection of a semi-public release model was based on best efforts to mitigate risk to participants while maximizing opportunities for data sharing and reuse. We decided that a public release model would not be appropriate given that the data sets was derived from a vulnerable pediatric cohort in a low-income country. On the other hand, a non-public release would considerably reduce data sharing opportunities. For these reasons, a semi-public release has been the general model adopted by the Pediatric Sepsis CoLab for sharing and distributing data.

## Step 2: Classify the variables

The second step involves determining which of the collected variables in the data sets contain identifying information. Classification of variables as identifiers was established after consideration of three conditions: replicability, distinguishability, and knowability (Table 1) [20]. Variables that fulfilled at least one of the three conditions were further classified as direct or quasi-identifiers using a decision tree (Fig 2). Variables that uniquely identified an individual or have been classified by the HIPAA as a direct identifier were treated as direct identifiers [15]. For the remaining variables, usefulness in data analysis was considered [20]. Variables defined as useful are those that are essential to the scientific objectives of the study.

To exemplify this, consider a variable containing information about the sex of a participant. This variable would fulfill conditions of replicability, as participant sex will likely not change over time and of knowability, as sex is common information likely knowable by an

**Table 1. Suggested conditions for classifying variables as identifiers.**

| Condition | Description |
|---|---|
| Replicability | The variable must be sufficiently stable over time so that the values will occur consistently in relation to the data subject. If a field value is not replicable, it will be challenging for an adversary to use that information to re-identify an individual. |
| Distinguishability | The variable must have sufficient variability to distinguish among individuals in a data sets. |
| Knowability | An adversary must know the identifiers about the data subject to re-identify them. This assumes the adversary is an acquaintance of a data subject. If a variable is not knowable by an adversary, it cannot be used to launch a re-identification attack on the data. |

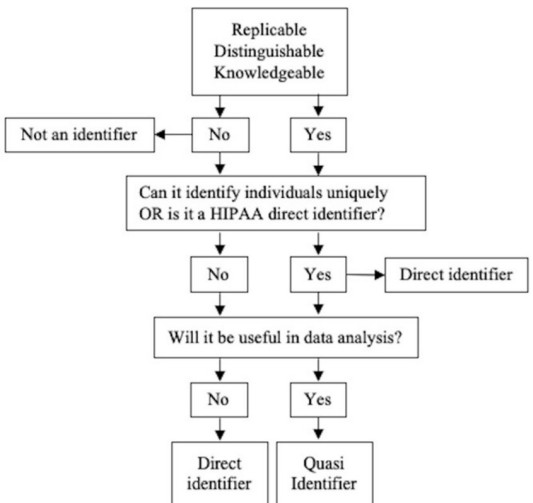

**Fig 2. Decision tree for classifying variables as identifiers.** Adapted from the Committee on Strategies for Responsible Sharing of Clinical Trial Data [20].

acquaintance (Table 1). This variable is not distinguishable as there were only two response options, male and female, that were approximately equally prevalent among participants in the data sets. Next, ask if this variable can uniquely identify an individual and check if it is a HIPAA direct identifier (Fig 2). It is not possible to directly identify an individual based only on knowledge of their sex, and it is not in the list of direct identifiers outlined by HIPAA. Finally, ask if this variable will be useful in data analysis. The Smart Triage data set is used to develop prediction models, and sex is a predictor variable used in analysis. Thus, we determine sex to be a quasi-identifier. In another example, take a clinical sign such as wheezing. This variable would not remain stable over time, is a common symptom among individuals in the data set and thus does not have sufficient distinguishability, and an acquaintance is not very likely to know whether a child was wheezing at the time of presentation to the hospital. Thus, this variable, along with many clinical signs and symptoms, would not be considered an identifier.

The variable classification process was to be conducted independently by the two investigators most familiar with the data, and Cohen's Kappa was computed to measure degree of agreement [21]. If the value was above 0.8, consensus was assumed, and the two investigators met and resolved the classifications on which they had disagreements. If the Kappa threshold was not achieved, the process was to be repeated with the full group of investigators. The results were then circulated to the study co-investigators to review and approve the variable classifications. Finally, the results were reviewed by an expert in research ethics and privacy compliance.

## Step 3: Determine the re-identification risk threshold

For a data sets to be considered de-identified, the data risk must be sufficiently reduced so that it is less than or equal to the re-identification risk threshold. Determination of an acceptable re-identification risk threshold required an assessment of the extent to which the release of the data set would invade an individual's privacy [14]. Three factors were considered to rank the level of potential privacy invasion as low, medium, or high: (1) the sensitivity of the data (the greater the sensitivity of the data, the greater the invasion of privacy), (2) the potential injury to patients from an inappropriate disclosure (the greater the potential for injury, the greater the invasion of privacy), and (3) the appropriateness of consent for disclosing the data (the less

appropriate the consent, the greater the invasion of privacy) [20]. The rank of potential privacy invasion was translated to a re-identification risk threshold needed to ensure an acceptable level of risk (Table 2) [14].

## Step 4: Quantify data risk

The k-anonymity model was used to measure data risk. Each set of records with indistinguishable quasi-identifiers is called an equivalence class, of size k. The model assumes an upper bound of $1/k$ on the probability of re-identification for each individual record [16,22,23]. This probability applies under two conditions: (1) the adversary (an individual who is attempting to use the data for a nefarious purpose) knows someone in the real world and is trying to find the record that matches that individual, or (2) the adversary has selected a record in the data sets and is trying to find the identity of that person in the real world [20]. A data sets is k-anonymous if each of its records cannot be distinguished from at least k-1 records and the k value required to consider a data set de-identified can be derived by taking the reciprocal of the re-identification risk threshold (Table 2). In a semi-public release model, the data risk is equal to the maximum re-identification risk across all records with the assumption that there will be a re-identification attack [14].

## Step 5: De-identify the data

All de-identification procedures were performed using R (3.5.1) [24].

## Preparing quasi-identifiers

Quasi-identifiers were isolated and summarized, each in terms of the total number of response values and the number of distinct responses. Frequency tables were used to summarize categorical variables to assess prevalence of each distinct response value within a variable, and to help inform creation of generalization hierarchies. Quasi-identifiers with less than 10% of participant responses were removed from the de-identified data set. Those that were captured in other variables, or not relevant for analysis were also removed. Finally, variables containing duplicate information were merged. The remaining quasi-identifiers were labelled based on their function in the development of a logistic triage model as one of (a) predictor variable, (b) outcome variable, or (c) supplementary variable (not included in modeling). The labelled quasi-identifiers were then assigned a relative numerical ranking based on importance in modelling. Three rounds of generalization hierarchies were created for the categorical variables.

## De-identification of quasi-identifiers

De-identification was conducted using the R package sdcMicro, which enables application of statistical disclosure control methods to the data to decrease re-identification risk of the data [25]. A cycle that applied generalization, followed by suppression was used to determine the optimal de-identification model (Fig 3). The first generalization hierarchy was applied to

**Table 2. Acceptable risk thresholds for different levels of privacy invasion suggested by the Ontario Information and Privacy Commissioner.**

| Invasion of Privacy | Re-identification Risk Threshold | k-Anonymity Equivalent |
|---|---|---|
| Low | 0.1 | 10 |
| Medium | 0.075 | 15 |
| High | 0.05 | 20 |

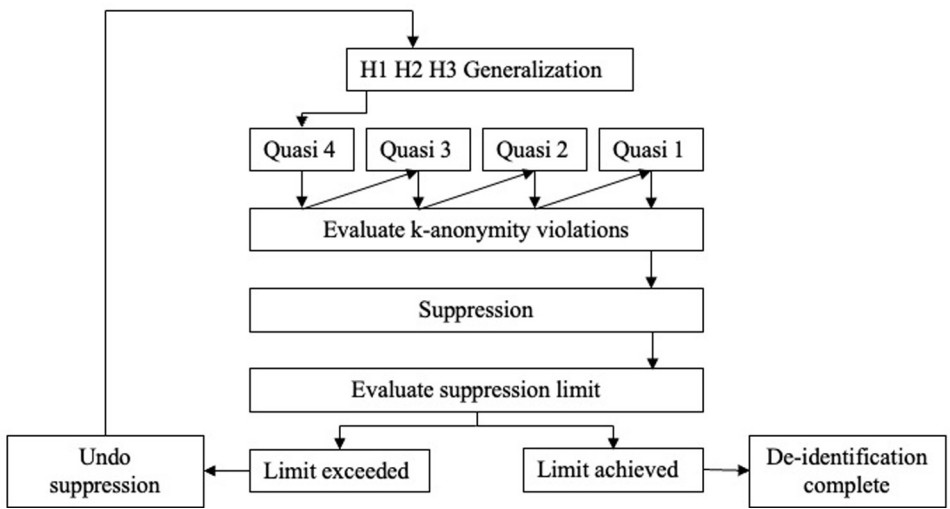

**Fig 3. De-identification schematic.** H1-H3 refer to consecutive generalization hierarchies. Numbers assigned to quasi-identifiers correspond to relative importance ranking.

applicable variables in a stepwise fashion, where the quasi-identifier with the lowest importance ranking was de-identified first. At each step, the number of records violating k-anonymity was evaluated. If k-anonymity violations remained after application of the first generalization hierarchy, suppression was applied to meet the requirement. The suppression algorithm was based on importance ranking of variables to prioritize preservation of modelling variables. It was decided that acceptable suppression limits were 5% and 10% for modelling and supplementary variables, respectively. If the number of suppressed values for a given variable exceeded the suppression limit, the above process was undone and repeated using generalization hierarchies two, and if needed, three. The model that requires the least amount of generalization for suppressions to be contained within the limit would be considered optimal.

## Step 6: Test data utility

The purpose of this data sets is to develop rapid triage models based on need for hospital admission, and thus assessment focused on evaluating model integrity. Missing data and the distribution of response values among quasi-identifiers were compared before and after suppression. Univariate logistic regression with 10-fold cross validation was applied to quasi-identifiers labelled as predictor variables on both the original and de-identified data sets. The outcome measure for modelling was defined as a positive response on either the admitted or readmitted variable. The magnitude and direction of the regression coefficients, as well as the significance of each variable in predicting the outcome were compared.

## Results

### Classification of variables

There were 22 (9.1%) disagreements found among the two independent variable classifications; however, the Cohen's Kappa test demonstrated an acceptable degree of similarity (k = 0.83). The classification consensus revealed 15 direct identifiers, 53 quasi-identifiers, and 173 non-identifying variables (S2 Appendix). In accordance with the Safe Harbour guidelines outlined by HIPAA, all direct identifiers were removed from the de-identified data set [15].

## Data risk and re-identification risk threshold

Data were collected from children presenting with an acute illness. Health information from these children can be considered sensitive, however the risk of potential injury from disclosure was considered to be low. Due to the absence of electronic health records within LMIC health-care systems, adversaries would have very limited public data sources. At Jinja Hospital in particular, patient information is hand recorded in notebooks and stored only for admitted children, making data sharing with public registries unfeasible. Additionally, informed consent approving disclosure of de-identified health information for the purpose of data sharing was obtained prior to enrollment. With consideration of the above information, the level of privacy invasion was ranked as medium, which corresponds to a re-identification risk threshold of 0.075 or a k-anonymity equivalent of 15 [14]. The data risk, a reflection of the maximal re-identification risk across all records, was found to be 100%. This means that at least one record in the Smart Triage data sets had a unique set of quasi-identifier values, different from any other record in the data sets (k = 1). The goal of the de-identification was thus to apply generalization and suppression until k = 15, meaning all records have at least 14 other records with identical quasi-identifier values.

## Quasi-identifiers

Of the 53 quasi-identifiers, 45 were removed from the de-identified data sets (S2 Appendix). Many quasi-identifiers (N = 24) were removed due to having too few participant responses to confer indistinguishability in the data sets (S3 Appendix) (Table 3). Some variables were merged with another variable or removed as the information was captured elsewhere (N = 11). For example, date of admission recorded by the study team and date of admission reported by caregiver were merged into a single variable. Further, month of birth and year of birth variables were removed as information was captured in the calculated age variable. Variables were also removed if they contained more than 100 unique response values with few counts each (N = 7), or if the information was not reliable (N = 2) or relevant for data analysis (N = 1). There were 8 remaining quasi-identifiers that required further de-identification (Table 4). Generalization hierarchies were created for the four categorical quasi-identifiers (Table 5).

## De-identification results

Results indicated that the optimal model for de-identification of this data sets was use of the second generalization hierarchy, followed by suppression. After applying the second generalization hierarchy, there remained 384 (21.9%) records violating 15-anonymity (Table 6). The amount of suppression required to address these violations were well within the 5% limit for modelling variables, however 'district', which had 272 (15.5%) suppressions, exceeded the 10% limit for supplementary variables (Table 7). This variable continued to exceed the limit even after application of the third generalization hierarchy as 235 (13.4) suppressions were required.

**Table 3. Reasons for removal of quasi-identifiers from the de-identified data sets (N = 45).**

| Reason for Removal | N |
|---|---|
| Too few data collected (<10% of participant responses, or <1% of positive responses). | 24 |
| Information captured in another, less identifying variable. | 7 |
| Contains sensitive, re-identifying information (>100 unique responses). | 7 |
| Duplicate variable to be merged with the original variable. | 4 |
| Self-reported and unlikely to be reliable or useful in data analysis. | 2 |
| Not relevant to data analysis. | 1 |

**Table 4. Summary of quasi-identifiers requiring further de-identification.**

| Variable (type) | Rank | Values | Frequency (%) | Missing (%) |
|---|---|---|---|---|
| Admitted (outcome) | 1 | 1.No<br>2.Yes | 1139 (76.5)<br>411 (23.5) | 14 (0.8) |
| Readmitted (outcome) | 2 | 1.No<br>2.Yes | 1672 (97.6)<br>41 (2.4) | 51 (2.9) |
| Age in years (predictor) | 3 | Mean (sd): 24.3 (28.6)<br>min ≤ med ≤ max:<br>0 ≤ 13.8 ≤ 208.4<br>IQR (CV): 23 (1.2) | 620 distinct values | 8 (0.5) |
| Sex (predictor) | 4 | 1.Male<br>2.Female | 849 (48.3)<br>908 (51.7) | 7 (0.4) |
| Length of stay in days (predictor) | 5 | 1. 0<br>2. 1<br>3. 2<br>4. 3<br>5. 4<br>6. 5<br>7. 6<br>8. 7<br>9. 8<br>10. 9<br>11. 10<br>12. 11<br>13. 12<br>14. 13<br>15. 14<br>16. 15<br>17. 24<br>18. 28 | 1359 (77.8)<br>37 (2.1)<br>84 (4.8)<br>72 (4.1)<br>79 (4.5)<br>40 (2.3)<br>22 (1.3)<br>20 (1.1)<br>8 (0.5)<br>5 (0.3)<br>6 (0.3)<br>3 (0.2)<br>1 (0.1)<br>2 (0.1)<br>5 (0.3)<br>1 (0.1)<br>1 (0.1)<br>1 (0.1) | 18 (1.0) |
| Admission diagnosis (supplementary) | 6 | 1. Malaria<br>2. Pneumonia<br>3. Bronchiolitis<br>4. URTI (cold, flu, etc.)<br>5. Reactive airway disease<br>6. Gastroenteritis/diarrhoea<br>7. HIV/AIDS<br>8. CNS infections<br>9. Sepsis<br>10. Neonatal Sepsis<br>11. Other<br>12. Not applicable (wasn't admitted) | 150 (8.6)<br>42 (2.4)<br>2 (0.1)<br>2 (0.1)<br>14 (0.8)<br>7 (0.4)<br>1 (0.1)<br>1 (0.1)<br>74 (4.3)<br>6 (0.3)<br>98 (5.6)<br>1339 (77.1) | 28 (1.6) |
| Urgent referral (supplementary) | 7 | 1.No<br>2.Yes | 1651 (94.3)<br>99 (5.7) | 14 (0.8) |
| District (supplementary) | 8 | 1. Buikwe<br>2. Buvuma<br>3. Bugiri<br>4. Iganga<br>5. Jinja<br>6. Kamuli<br>7. Kayunga<br>8. Luuka<br>9. Mayuge<br>10. Namutumba<br>11. Other | 466 (26.5)<br>4 (0.2)<br>1 (0.1)<br>12 (0.7)<br>1151 (65.4)<br>22 (1.2)<br>19 (1.1)<br>12 (0.7)<br>46 (2.6)<br>2 (0.1)<br>25 (1.5) | 4 (0.2) |

sd, standard deviation; min, minimum; med, median; max, maximum; IQR, interquartile range; CV, coefficient of variation.

**Table 5. Generalization hierarchies for categorical quasi-identifiers.**

| Variable | Round 1 | Round 2 | Round 3 |
|---|---|---|---|
| Age | 5-month interval; 60+ | 12-month interval; 60+ | [0–12), [12–24), [24,36), [36,60); 60+ months |
| Length of stay | 1. 0 days<br>2. 1–2 days<br>3. 3–4 days<br>4. 5–6 days<br>5. 7+ days | 1. 0 days<br>2. 1–6 days<br>3. 7+ days | 1. 0 days<br>2. 1+ days |
| Admission diagnosis | 1. Malaria<br>2. Pneumonia and RTIs (2–5)<br>3. Sepsis (9,10)<br>4. Gastroenteritis/diarrhoea<br>5. Other (7,8,11) | 1. Malaria<br>2. Pneumonia and RTIs (2–5)<br>3. Sepsis (9,10)<br>4. Other (6–8,11) | 1. Malaria<br>2. Sepsis (9, 10)<br>3. Other (2–8,11) |
| District | 1. Jinja<br>2. Buikwe<br>3. Other | 1. Jinja<br>2. Other | 1. Jinja<br>2. Other |

Since suppression limits for modelling variables were met using the second generalization hierarchy, it was deemed impractical to further generalize variables in exchange for a small reduction in the number of required suppressions (Table 7).

## Data utility

Following suppression, prevalence of missing values continued to be minimal among modelling variables (Table 7). In addition, the distribution of response values for modelling variables pre- and post-suppression were within 1% of each other (Table 8). Considerable data loss was evident among supplementary variables and the distribution of response values pre-and post-suppression were variable (Tables 7 and 8).

Univariate logistic regression demonstrated that associations between the three predictor variables and outcomes were similar pre-and-post data de-identification (Table 9). In both cases, males were significantly less likely to have a positive admission outcome compared to females. Pre-and-post de-identification odds ratios showed overlapping confidence intervals. Length of stay was not significantly associated with the positive admission outcome at any time. Finally, both in the original and de-identified data set, increasing age was associated with a decrease in the odds of having a positive admission outcome. This association was significant in the original data set, but only for children aged three and older in the de-identified data set where age was transformed into categorical bins.

## Discussion

### Summary

We have proposed a standardized framework for the de-identification of data generated from cohort studies in children in a LMIC. In an effort to balance protection of patient privacy and

**Table 6. K-anonymity violations at compounding steps of generalization in order of increasing variable importance for the three hierarchical rounds, N (%).**

| Generalization Hierarchy: | No de-identification | +District | +Admission diagnosis | +Length of hospitalization | +Age |
|---|---|---|---|---|---|
| **Round 1** | 1750 (100.0) | 1749 (99.9) | 1749 (99.9) | 1749 (99.9) | 649 (37.1) |
| **Round 2** | 1750 (100.0) | 1749 (99.9) | 1749 (99.9) | 1749 (99.9) | 384 (21.9) |
| **Round 3** | 1750 (100.0) | 1749 (99.9) | 1749 (99.9) | 1749 (99.9) | 344 (19.7) |

This table demonstrates the number of records violating k-anonymity at each step of each round of generalization. The "+" sign indicates that generalization has been applied for this variable.

**Table 7. Records per variable suppressed to meet 15-anonymity requirement after each round of generalization, N (%).**

| Variable | Post-Round 1 | Post-Round 2 | Post-Round 3 |
|---|---|---|---|
| Admitted (outcome) | 1 (0.1) | 0 (0.0) | 0 (0.0) |
| Readmitted (outcome) | 2 (0.1) | 3 (0.2) | 2 (0.1) |
| Age (predictor) | 30 (1.7) | 18 (1.0) | 16 (0.9) |
| Sex (predictor) | 76 (4.3) | 32 (1.8) | 18 (1.0) |
| Length of stay (predictor) | 137 (7.8) | 39 (2.3) | 42 (2.4) |
| Admission diagnosis (supplementary) | 269 (15.4) | 115 (6.6) | 80 (4.6) |
| Urgent referral (supplementary) | 267 (15.3) | 161 (9.2) | 121 (6.9) |
| District (supplementary) | 456 (26.1) | 272 (15.5) | 235 (13.4) |

data integrity, direct identifiers were removed from the data sets and a de-identification model using generalization followed by suppression was applied to quasi-identifiers. The utility of the de-identified data was demonstrated using a typical clinical regression example. Use of the R package sdcMicro allowed for flexibility in manipulating variables, and ease of monitoring and evaluating de-identification progress at each step of each generalization-suppression combination [25].

The risk of re-identification of individual participants would be unacceptable with the suppression of direct identifiers alone. Statistical approaches to de-identification are recommended for mitigating re-identification risk associated with quasi-identifiers [14]. Two widely accepted privacy models are k-anonymity and differential privacy [26]. K-anonymity is used to prevent re-identification of individuals made possible by record linking attacks while the differential privacy provides a probabilistic guarantee that the inclusion of an individual in a data set will not alter the outcome of a query to that data sets [27]. There are many trade-offs between these techniques [28,29]. In the case of record-level data release, applying differential privacy would require employing a large amount of noise to obtain a meaningful privacy guarantee. As a result, the analytical utility of the output would be poor [30,31]. Thus, k-anonymity was deemed preferable for this microdata set.

Numerous methods have been proposed for reducing the risk of re-identification. There are two broad types of statistical disclosure control techniques that can be used to satisfy the parameters of the selected privacy model: i) non-perturbative techniques, such as generalization and local suppression, which suppress or reduce the detail without altering the original data, and ii) perturbative techniques, such as adding noise, post-randomization method, or micro-aggregating and shuffling, which distort the original data sets before release [32]. When dealing with health data, non-perturbative methods are favoured because the truthfulness of the data is preserved and the impact on data analysis is more easily evaluated [33].

In the described framework we chose to use the k-anonymity model with a combination of generalization and suppression. The focus of de-identification was to maximize data integrity in the context of the development and validation of risk prediction models while firmly protecting patient privacy. Thus, we favoured a de-identification scheme involving generalizations that satisfy k-anonymity with minimal suppression of key modelling variables, while also

**Table 8. Distribution of response values for variables pre-and-post suppression.**

| Variable | Pre-Suppression, N (%) | Post-Suppression, N (%) |
|---|---|---|
| **Admitted (outcome)** | | |
| Yes | 411 (23.49) | 411 (23.49) |
| No | 1339 (75.61) | 1339 (75.61) |
| **Readmitted (outcome)** | | |
| Yes | 41 (2.39) | 38 (2.22) |
| No | 1671 (97.61) | 1671 (97.78) |
| **Age in years (predictor)** | | |
| 0 | 787 (45.00) | 786 (45.41) |
| 1 | 439 (25.10) | 439 (25.36) |
| 2 | 157 (8.98) | 151 (8.72) |
| 3 | 111 (6.35) | 108 (6.24) |
| 4 | 72 (4.12) | 68 (3.93) |
| 5+ | 183 (10.46) | 179 (10.43) |
| **Sex (predictor)** | | |
| Female | 846 (48.34) | 830 (48.31) |
| Male | 904 (51.66) | 888 (51.69) |
| **Length of stay in days (predictor)** | | |
| 0 | 1359 (77.84) | 1340 (78.50) |
| 1–6 | 334 (19.13) | 327 (19.16) |
| 7+ | 53 (3.04) | 40 (2.34) |
| **Admission diagnosis (supplementary)** | | |
| Malaria | 150 (8.64) | 122 (7.53) |
| Pneumonia and other RTIs | 47 (2.71) | 26 (1.60) |
| Sepsis | 80 (4.61) | 60 (3.70) |
| Other | 120 (6.91) | 74 (4.57) |
| Not applicable | 1339 (77.14) | 1339 (82.60) |
| **Urgent referral (supplementary)** | | |
| Yes | 99 (5.66) | 9 (0.57) |
| No | 1649 (94.34) | 1578 (99.43) |
| **District (supplementary)** | | |
| Jinja | 1148 (65.60) | 1041 (70.43) |
| Other | 602 (34.40) | 437 (29.57) |

minimizing generalizations at the expense of suppressing supplementary variables. To further increase data utility, generalization was applied to quasi-identifiers in a stepwise fashion in order of increasing importance in modelling and data was evaluated at each step to determine the earliest point of k-anonymity achievement. Importance ranking of quasi-identifiers was also applied in suppression to minimize suppression of key modelling variables.

## Data utility

The Smart Triage data sets was generated to inform development of rapid triage models. Thus, it was important to maximize integrity of the modelling variables and minimize distortions of associations between predictor and outcome variables. The modelling outcomes, admitted and readmitted, had a total of 0 (0.0%) and 3 (0.2%) suppressions respectively in the de-identified data set. Suppressions among predictor variables were also minimal with 18 (1.0%) for age, 32 (1.8%) for sex, and 42 (2.4) for length of stay. For these variables, the

**Table 9. Results from univariate logistic regression pre-and-post de-identification.**

| Variable | Regression Coefficient | p-value | Odds Ratio (95% CI) |
|---|---|---|---|
| **Original Data** | | | |
| **Sex (Male)** | -0.118 | <0.05 | 0.844 (0.792, 0.985) |
| **Age** | -0.107 | <0.05 | 0.988 (0.812, 0.998) |
| **Length of stay** | 8.14 | 0.971 | |
| **De-identified data** | | | |
| **Sex (Male)** | -0.198 | <0.05 | 1.20 (1.07, 1.34) |
| **Age in years (Baseline = 0)** | | | |
| *1* | -0.025 | 0.671 | |
| *2* | -0.045 | 0.588 | |
| *3* | -0.121 | <0.05 | 0.886 (0.800, 0.984) |
| *4* | -0.026 | <0.05 | 0.975 (0.877, 1.09) |
| *≥5* | -0.054 | <0.05 | 0.947 (0.849, 1.06) |
| **Length of stay in days (Baseline = 0)** | | | |
| 1–6 | 8.760 | 0.970 | |
| ≥7 | 3.359 | 0.989 | |

CI, Confidence Interval

distribution of response values before and after suppression are within 1% of each other. This suggests that the integrity of these variables remains intact. Univariate regression revealed the sex and length of stay predictors to have similar associations to the outcome in the original and de-identified data sets. Age, however, was found to be significantly associated to the admissions outcome in the original data set, but only for children aged three and older in the de-identified data set. This resulted from the transformation of age as a continuous variable in the original data sets to a categorical variable in the de-identified data set. These transformations are generally not advised due to the risk that variables may lose predictive value and that associations with the modelling outcome may be distorted [34]. Generalization was applied nonetheless in the interest of protecting patient privacy and to achieve k-anonymity with reasonable suppression. The supplementary variables were generously suppressed to maximize integrity of the modelling variables. This was acceptable as it allowed for at least some supplementary information to be retained, but not at the expense of compromising the development of prediction models.

## Limitations

Though many quasi-identifiers were removed from the data sets, the quantity of collected data for these variables was not sufficient for meaningful analysis and the rarity in positive responses presented a high risk of privacy invasion (Table 3). The low quantity of responses for many of these quasi-identifiers can be attributed to the limited applicability to a small subset of participants. Using branching survey logic, these variables were not collected for most participants. Alternatively, some variables by nature were associated with few occurrences. For example, the removal of mortality as an outcome was inevitable due to the small number of cases with this outcome (N = 10) (S3 Appendix). There would also be a high likelihood of linking this outcome to other data sources. There does not appear to be a reasonable method to generalize or suppress this outcome that would reduce the risk of re-identification. Increasing the sample size and using multiple undisclosed locations would allow us to reduce the risk of re-identification in the future.

A second limitation is that integrity of the supplementary variables was significantly reduced to preserve the modelling variables. While this was necessary in order to maximize data integrity for the primary purpose of prediction modelling, it does reduce opportunities for secondary analysis where specific focus may be placed on suppressed data. For example, researchers wishing to do a deeper dive into urgent referrals should be aware that this de-identified data set may not be suitable. This could highlight a need for a new collaborative agreement between investigators so that the original data set can be used for such analyses.

The generalization of age was a particular concern. The physiology of children changes rapidly following birth with rapid transitions occurring in the first year or two of life [35]. The loss of the number of months of age in these children is potentially limiting. The granularity in the data was lost due to generalization. This was required because of the small number of children in the data set of the same age. The fact that the date of birth had been generalized to age at the time of triage had already provided a significant degree of risk reduction that was not considered. While age is typically considered to be a quasi-identifier, in a study that has a duration of many months or years, converting the date of admission and date of birth provides generalization if these dates are suppressed and the date of admission could be anywhere within the study period. The same would be true of other time intervals such as length of stay. Further, as all variables classified as dates in our study were converted into time intervals, we provided limited guidance on how to de-identify these types of variables using generalization and suppression.

Another limitation was the difficulty in assessing data utility. In a complex data sets generated to develop prediction models, where variables are of varying type and held at different levels of importance, there are no broadly accepted metrics that can be applied to judge the results [36]. Additionally, no existing de-identification algorithm provides both perfect privacy protection and perfect analytic utility. Thus, selection of an acceptable re-identification risk threshold to balance the probability of re-identification with the amount of distortion applied to the data was based on subjective assessment of privacy risk. It is impossible to anticipate all potential unethical future uses of data that could lead to significant harms such as discrimination or stigmatization of individuals or groups. It is for this reason that the standardized approach proposed here should be understood as one tool used to manage privacy risks within a greater holistic data governance framework.

## Conclusion

We presented a standardized framework for de-identification of health data using statistical disclosure control methods. This was in effort to reduce barriers to Open Data sharing by demonstrating the possibility of maintaining data integrity while protecting patient privacy.

## Supporting information

**S1 Appendix. Functional definitions of key terms used in data de-identification.**
(PDF)

**S2 Appendix. List and classification of variables in the Smart Triage data sets.**
(XLSX)

**S3 Appendix. Quasi-identifier summary (N = 53).**
(XLSX)

## Acknowledgments

We thank Jessica Trawin and the Pediatric Sepsis Data CoLaboratory. We also thank WALIMU, the administration and staff of Jinja Regional Referral Hospital, and participants and caregivers for their time and dedication to Smart Triage.

## Author Contributions

**Conceptualization:** Alishah Mawji, Holly Longstaff, J. Mark Ansermino.

**Data curation:** Alishah Mawji, Jessica Trawin, Dustin Dunsmuir, Clare Komugisha, J. Mark Ansermino.

**Formal analysis:** Alishah Mawji.

**Funding acquisition:** J. Mark Ansermino.

**Investigation:** Matthew O. Wiens, Samuel Akech, Abner Tagoola, Niranjan Kissoon.

**Methodology:** Alishah Mawji.

**Software:** Dustin Dunsmuir.

**Supervision:** Holly Longstaff, Matthew O. Wiens, Samuel Akech, Niranjan Kissoon, J. Mark Ansermino.

**Writing – original draft:** Alishah Mawji.

**Writing – review & editing:** Holly Longstaff, Jessica Trawin, Dustin Dunsmuir, Clare Komugisha, Stefanie K. Novakowski, Matthew O. Wiens, Samuel Akech, Niranjan Kissoon, J. Mark Ansermino.

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
