## [Decision Letter · Decision Letter 0]

31 May 2022

PDIG-D-22-00081

A proposed de-identification framework for a cohort of children presenting at a health facility in Uganda

PLOS Digital Health

Dear Dr. Mawji,

Thank you for submitting your manuscript to PLOS Digital Health. After careful consideration, we feel that it has merit but does not fully meet PLOS Digital Health's publication criteria as it currently stands. Therefore, we invite you to submit a revised version of the manuscript that addresses the points raised during the review process.

The reviewers comments were generally positive and we are in principle interested in publishing the manuscript subject to your response to the reviewers.

Please submit your revised manuscript by . If you will need more time than this to complete your revisions, please reply to this message or contact the journal office at digitalhealth@plos.org. Please include the following items when submitting your revised manuscript:

We look forward to receiving your revised manuscript.

Kind regards,

Jasmit Shah

Guest Editor

PLOS Digital Health

Hamish Fraser

Section Editor 

PLOS Digital Health

Journal Requirements:

State the initials, alongside each funding source, of each author to receive each grant.

2. Please update the completed 'Competing Interests' statement. Please declare all competing interests beginning with the statement “I have read the journal's policy and the authors of this manuscript have the following competing interests:”.

Additional Editor Comments (if provided):

Reviewers' comments:

Reviewer's Responses to Questions

**Comments to the Author**

1. Does this manuscript meet PLOS Digital Health’s publication criteria? Is the manuscript technically sound, and do the data support the conclusions? The manuscript must describe methodologically and ethically rigorous research with conclusions that are appropriately drawn based on the data presented.

Reviewer #1: Yes

Reviewer #2: Yes

Reviewer #3: Yes

Reviewer #4: Yes

2. Has the statistical analysis been performed appropriately and rigorously?

Reviewer #1: Yes

Reviewer #2: I don't know

Reviewer #3: Yes

Reviewer #4: Yes

3. Have the authors made all data underlying the findings in their manuscript fully available (please refer to the Data Availability Statement at the start of the manuscript PDF file)?

Reviewer #1: Yes

Reviewer #2: Yes

Reviewer #3: Yes

Reviewer #4: Yes

4. Is the manuscript presented in an intelligible fashion and written in standard English?

Reviewer #1: Yes

Reviewer #2: Yes

Reviewer #3: Yes

Reviewer #4: Yes

5. Review Comments to the Author

Reviewer #1: Thank you for conducting the study related to the protection of research participant data. Since privacy is extremely important nowadays, especially the research contained enormous personal information. The manuscript is well written and organized in detail. I have no further comment on the manuscript. Good luck to you.

Reviewer #2: This report includes a detailed description of a method used to protect identifiable information in research. The methods described in this study contribute new knowledge and could have public health implications by improving our knowledge on how to optimize data access whilst protecting identifiable health information, potentially fostering easier collaboration and hence data sharing. Further, by advocating for and outlining methods for implementing de-identification approaches, the findings have the ability to alleviate common concerns about data security and data protection to enhance public confidence in the scientific process. In summary, the report could alleviate barriers to conducting future research that will help to inform public health decisions. 

However, some sections lacked clarity and require additional information for the reader to replicate the method. It is also unclear whether the approach described in the current study can be applied to many types of datasets or research questions, and how much it can affect the replicability of statistical analyses. Perhaps including a second example, as well as sharing the R code used to generate the de-identified dataset would greatly improve this paper.

Reviewer #3: General comments

The manuscript is well written and all sub sections are clear to read. 

The manuscript adequately describe the methods used.

Comment. Under limitations, Line 480 the sentence “Though it may appear concerning that many quasi-identifiers…” This sentence is not very clear. It is recommended to delete the word “concerning”.

Thank you

Reviewer #4: Question 3 - While answer is Yes, the authors indicated that the data studied and code are available on the Pediatric Sepsis CoLab Dataverse and provided the link and contacts. Data is available subject to an application that meets the ethical and governance requirements.

Additional comments:

This is a masterpiece paper, with the proposed methodology for data de-identification having a significant potential to positively contributing to research, even beyond the health sector. Any researcher who wants to comply with funder or publisher requests to make data open can easily follow the step-by-step guidelines proposed.

6. PLOS authors have the option to publish the peer review history of their article (what does this mean?). If published, this will include your full peer review and any attached files.

**Do you want your identity to be public for this peer review?** For information about this choice, including consent withdrawal, please see our Privacy Policy.

Reviewer #1: Yes: Yuen-ling LEUNG

Reviewer #2: No

Reviewer #3: No

Reviewer #4: Yes: Wellington Mushayi

---

## [Editor Report · Decision Letter 1]

8 Jul 2022

A proposed de-identification framework for a cohort of children presenting at a health facility in Uganda

PDIG-D-22-00081R1

Dear Miss Mawji,

We are pleased to inform you that your manuscript 'A proposed de-identification framework for a cohort of children presenting at a health facility in Uganda' has been provisionally accepted for publication in PLOS Digital Health.

Best regards,

Jasmit Shah

Guest Editor

PLOS Digital Health